# Ultrasound imaging identifies life history variation in resident Cutthroat Trout

**Kellie J. Carim**[1,2]*, **Scott Relyea**[3], **Craig Barfoot**[4], **Lisa A. Eby**[2], **John A. Kronenberger**[1], **Andrew R. Whiteley**[2], **Beau Larkin**[5]

**1** National Genomics Center for Wildlife and Fish Conservation, U.S. Forest Service, Rocky Mountain Research Station, Missoula, Montana, United States of America, **2** Wildlife Biology Program, University of Montana, Missoula, Montana, United States of America, **3** Sekokini Springs Hatchery, Montana Fish Wildlife and Parks, Montana, United States of America, **4** Confederated Salish and Kootenai Tribes, Pablo, Montana, United States of America, **5** MPG Ranch, Condon, Montana, United States of America

* kellie.carim@usda.gov

**Data Availability Statement:** All relevant data are within the manuscript and its Supporting information files.

## Abstract

Human activities that fragment fish habitat have isolated inland salmonid populations. This isolation is associated with loss of migratory life histories and declines in population density and abundance. Isolated populations exhibiting only resident life histories may be more likely to persist if individuals can increase lifetime reproductive success by maturing at smaller sizes or earlier ages. Therefore, accurate estimates of age and size at maturity across resident salmonid populations would improve estimates of population viability. Commonly used methods for assessing maturity such as dissection, endoscopy and hormone analysis are invasive and may disturb vulnerable populations. Ultrasound imaging is a non-invasive method that has been used to measure reproductive status across fish taxa. However, little research has assessed the accuracy of ultrasound for determining maturation status of small-bodied fish, or reproductive potential early in a species' reproductive cycle. To address these knowledge gaps, we tested whether ultrasound imaging could be used to identify maturing female Westslope Cutthroat Trout (*Oncorhynchus clarkii lewisi*). Our methods were accurate at identifying maturing females reared in a hatchery setting up to eight months prior to spawning, with error rates ≤ 4.0%; accuracy was greater for larger fish. We also imaged fish in a field setting to examine variation in the size of maturing females among six wild, resident populations of Westslope Cutthroat Trout in western Montana. The median size of maturing females varied significantly across populations. We observed oocyte development in females as small as 109 mm, which is smaller than previously documented for this species. Methods tested in this study will allow researchers and managers to collect information on reproductive status of small-bodied salmonids without disrupting fish during the breeding season. This information can help elucidate life history traits that promote persistence of isolated salmonid populations.

## Introduction

Human activities that degrade and fragment freshwater habitats reduce the abundance, distribution, and persistence of stream-dwelling fishes [1,2]. For salmonid species, one consequence

**Funding:** Funding for this research was provided by MPG Ranch to KJC and BL. Employees of MPG Ranch assisted in data collection on MPG Ranch property.

**Competing interests:** The authors have declared that no competing interests exist.

of these anthropogenic influences is population isolation and subsequent loss of migratory life histories. In populations with partial migratory life histories, larger migratory females lay more eggs than their smaller resident counterparts [3–6]. The loss of migratory life histories as a result of human activities has been associated with lower average fecundity in populations as well as decreased fish density and abundance [7,8]. Furthermore, isolated populations may be more vulnerable to extirpation because migrants are unable to augment populations or recolonize vacant habitat [9]. As fragmentation of freshwater habitat increases across the landscape, conservation practitioners are increasingly challenged with managing inland salmonids in the absence of migratory life histories. Therefore, understanding the mechanisms that promote persistence of populations with primarily resident life-histories will be critical for effective management and conservation.

One mechanism that may increase persistence of isolated salmonid populations is maturation at younger ages and smaller sizes [10,11]. Maturation at a smaller size is often indicative of maturation at a younger age, and would increase fitness if it increases the number of reproductive events in an individual's lifetime [12]. Substantial variation in average size at maturity occurs both within and among salmonid populations [13–15]. The factors influencing size at maturity are diverse, and have been linked to both local and regional habitat variables [15–19]. For example, increasing latitude is associated with larger size at maturity [16,19]. On a localized scale, inter- and intra-specific competition, lower stream discharge, temperature, and food quality are all associated with smaller size at maturity [14,15,17,18]. Given this variation, measuring reproductive status of salmonids requires methods that are accurate across individuals of different sizes.

Several methods exist for assessing reproductive status of fishes. Dissection is the most definitive method; however, lethal sampling is not well-suited for populations of conservation concern. Common non-lethal methods include endoscopy and blood plasma analyses. Endoscopy allows a surveyor to visually assess gonads using a small camera. Blood plasma may be analyzed for the presence and concentrations of hormones (i.e., various estrogens and androgens) or proteins (e.g., vitellogenin, a precursor yolk protein) involved in gonad development [e.g., 20, 21]. All of the methods above have been successfully used to determine sex and maturation status in large fishes such as such as sturgeon and sharks [e.g., 22–24]. However, in smaller fishes such as small resident salmonids, endoscopy is likely to be overly invasive and can be less accurate [25]. Furthermore, obtaining blood volumes needed for blood plasma analyses on small-bodied fishes may have harmful or even lethal consequences.

Ultrasound imaging is a non-invasive alternative that has been used broadly to determine sex and maturation in fish [26]. Studies using this tool have generally focused on fish that are ≥ 200 mm and up to 1–2 m in total length (TL), and have addressed questions related to sex, gonad size and spawn status during the spawning season [26]. In salmonids, ultrasound imaging has primarily been used on anadromous fish, and usually immediately before or during the spawning season [20,27–29]. Few studies have assessed the utility of ultrasound to determine maturity of smaller bodied fishes, such as resident salmonids. Additionally, imaging fish during the spawning season can be problematic for species that spawn in difficult sampling conditions like seasonal floods, and disrupting populations during this time may be detrimental to spawning success. If ultrasound imaging can be used to determine maturity of small-bodied fish before the spawning season, researchers will have a means of better understanding the connection between life history variation and population persistence in isolated populations of salmonid fish. One salmonid species for which these tools and information would be particularly useful is the Cutthroat Trout (*Oncorhynchus clarkii*).

Over the last century, many Cutthroat Trout populations have been isolated by habitat degradation and modification as well as the intentional construction of barriers to restrict spread

of invasive species. Such isolation has caused widespread elimination of fluvial life history forms, and has likely contributed to reducing Cutthroat Trout distribution throughout their historic range [30–33]. To manage and conserve these isolated populations, an understanding of the life history traits associated with persistence is crucial. However, many life history traits are challenging to directly measure because Cutthroat Trout spawn during spring flood events, typically in mountain headwater streams that may be difficult to access given spring snow cover. Moreover, sampling at this time could disrupt spawning behavior and reduce spawning success. These challenges have forced researchers modeling population persistence of salmonids to generalize limited information on size at maturity [10,34–36], possibly leading to inaccurate estimates of population viability [10]. Managers could therefore benefit from a reliable, less invasive means of assessing the reproductive status and size at maturity in salmonid populations such as Cutthroat Trout. If ultrasound imaging may be able to address these information gaps if it can accurately identify maturing fish prior to the spawning season.

To date, few published studies have applied ultrasound methods to fish more than 6 months pre-spawn or to fish as small as resident Cutthroat Trout [26], which can mature at sizes smaller than 140 mm TL [37–39]. Limited research on Coastal Cutthroat Trout (*O. c. clarkii*) indicates that oocyte development in maturing females begins more than eight months prior to spawning [21,40]. This suggests that it may be possible to determine reproductive status of Cutthroat Trout with ultrasound well before the spawning season. If ultrasound imaging can reliably assess reproductive status for small-bodied salmonids early in their reproductive cycle, it could be a valuable tool for measuring variation in life histories. This information, in turn, could be applied to models that elucidate how life history variation may influence population persistence [e.g., 10, 11, 41].

To better understand the use of ultrasound imaging to detect life history variation in resident populations of salmonid fish, we focused on two main research objectives:

1. Validate the use of ultrasound imaging to identify maturing female Westslope Cutthroat Trout (*O. c. lewisi*) outside the breeding season.

2. Use ultrasound imaging to elucidate variation in size of small-bodied, maturing females among resident Westslope Cutthroat Trout populations.

## Methods

Fish collection and handling in this study occurred under Montana Fish Wildlife and Parks permits SCP-16-2018 and SCP-33-2019, and animal care and use protocol 036-19LEECS-062419 approved by the University of Montana Institutional Animal Care and Use Committee.

### Hatchery fish surveys

Initial work on our first objective occurred at Sekokini Springs Hatchery in West Glacier, MT. Sekokini Springs Hatchery rears and spawns Westslope Cutthroat Trout to augment and restore wild populations in western Montana without creating a hatchery broodstock. All fish at Sekokini Springs Hatchery are sourced from wild populations, with collections targeting individuals between 130–220 mm TL. Upon entering the hatchery, each fish is implanted with a unique PIT tag to allow hatchery managers to track the health and reproductive development of individuals. Fish are reared in the hatchery until they reach maturity and are spawned (typically 2 years after entering the hatchery). Hatchery-reared fish in this study included fish sourced from three unfragmented streams in northwestern Montana: Danaher and Young's

Creeks, both tributaries to the South Fork Flathead River, and Sullivan Creek, a tributary to Hungry Horse Reservoir (Fig 1).

We surveyed hatchery-reared fish in the fall (October or November), winter (January), and early spring (March) prior to spawning in late spring (late May to early June) of 2018 and 2019. We did not attempt to survey the same individuals on repeated visits, although 211 individuals were surveyed on multiple visits by chance. These repeated surveys allowed us to track accuracy of ultrasound assessments over time within a spawning cycle. Results from ultrasound examinations were validated each year with spawning and post-spawning inventories in late spring. Hatchery-reared fish were classified as mature females if they expressed eggs and/or possessed an ovipositor. Fish were classified as mature male if they expressed milt and/or possessed a kype. A reliable genetic marker for sex identification of Westslope Cutthroat Trout did not exist at the time of this study. Therefore, we were unable to classify fish that did not display any of the sex characteristics above; sex for these individuals was considered "unknown".

### Wild fish surveys

Fish from Sekokini Springs Hatchery were typically larger than resident fish observed in wild populations. To determine whether ultrasound imaging could also be used on smaller fish (< 200 mm TL) we lethally sampled fish entrained in Gunderson Ditch, an unscreened irrigation ditch fed by West Revais Creek that is dewatered in summer months and located on the Flathead Indian Reservation in western MT (Fig 1). Fish were collected on 16 and 20 November 2017 (n = 30), and 5 November 2018 (n = 21), approximately 6–7 months prior to spawning. Although larger fish were encountered, field crews targeted fish between 100–150 mm TL for these surveys. Fish were euthanized with an overdose of eugenol in the form of FA-100 (a pharmaceutical preparation of 10% eugenol), measured for TL, stored individually in plastic bags, and frozen within 4 hours of capture. Fish remained frozen until ultrasound imaging occurred. Unfortunately, a reliable genetic marker for sex identification of Westslope Cutthroat Trout is not currently available. As a result, the presence of secondary sex characteristics was necessary to classify individuals as male or female. Sex of each fish was determined by dissection immediately after an individual's ultrasound examination. Fish with pronounced, milky testes were classified as mature male; fish with distinct and easily observed oocytes were classified as maturing females [40]. If gonads were poorly developed (e.g., oocytes were very small, misshapen, and/or hard to distinguish with the naked eye) or not easily observed during dissection, the individual was considered immature.

During dissection, saggital otoliths were removed from fish captured in Gunderson Ditch and used to determine the age of the smallest mature female. Age was assessed by visualizing annuli following methods outlined by Corsi et al. These data also were added to a larger dataset that includes length-at-age keys for other wild populations in this study [10].

Validation of our ultrasound imaging technique on trout < 150 mm TL from Gunderson Ditch enabled us to confidently survey five additional wild populations with minimal lethal sampling. Fish were surveyed from populations in the Swan River basin (Cooney Creek), lower Flathead River basin (Magpie Spring, Schley, and Yellow Bay Creeks) and the Bitterroot River basin (South Fork Davis Creek) in October of 2018 and 2019 (Fig 1). These surveys allowed us to address our second objective related to variation in size at maturity in female trout. We used backpack electrofishing techniques to collect Westslope Cutthroat Trout, targeting individuals > 100 mm TL to minimize stress on juvenile (immature) fish. We aimed to sample at least 50 fish per population; however, achieving this sample size was not always possible due to low fish densities in some populations. During surveys of wild populations, we

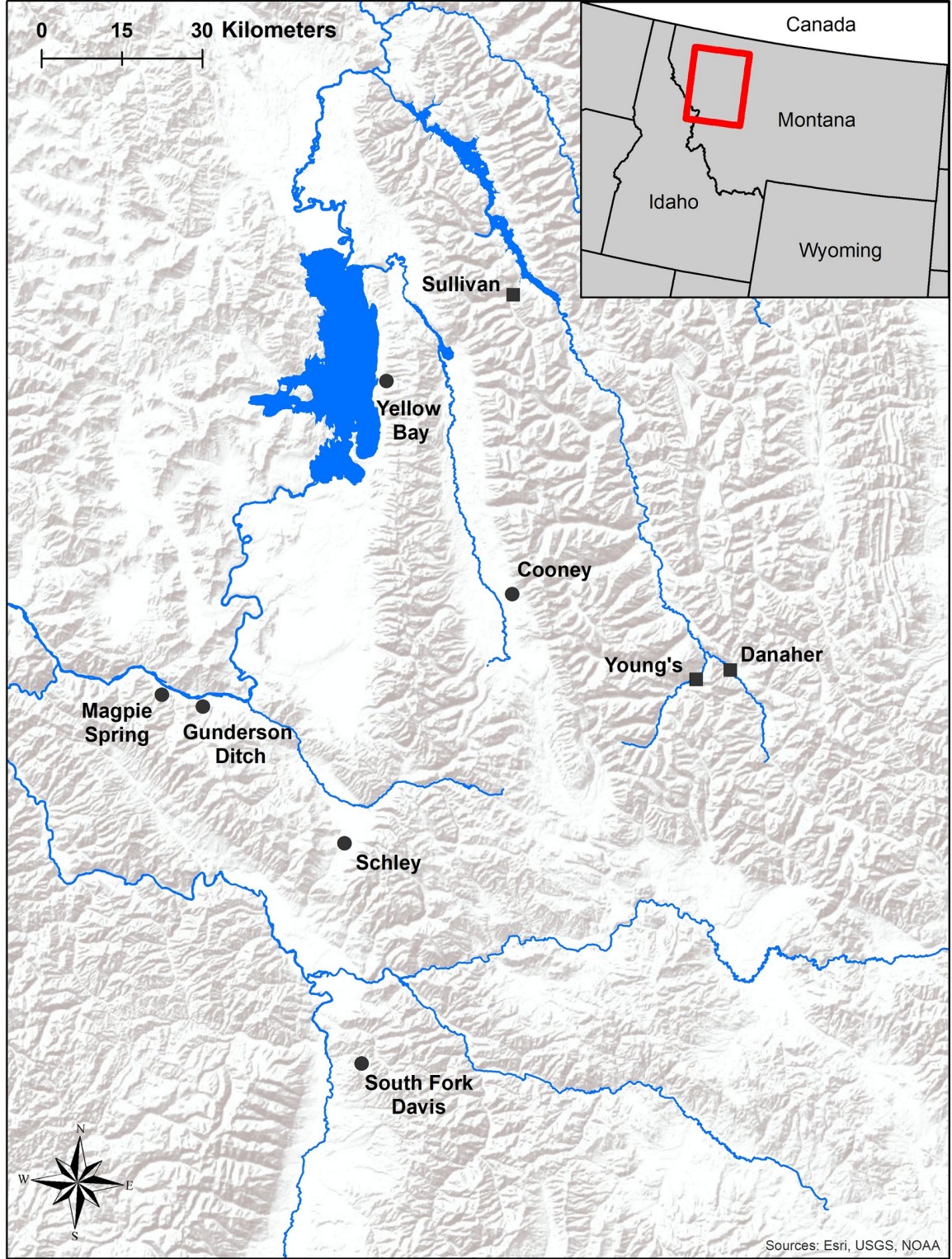

**Fig 1. Locations of source populations for fish raised at Sekokini Spring Hatchery (squares) and wild resident populations (circles) surveyed in this study.**

euthanized and dissected one to five fish per population to validate ultrasound examinations conducted in the field. This was useful for orienting technicians to ultrasound images, particularly when examining small fish.

## Ultrasound examination protocol

Immediately prior to imaging, live fish were sedated with FA-100 eugenol diluted in hatchery or stream water at a concentration of ~50–100 mg/L. Total length in mm (all fish) and PIT tag identification number (hatchery fish only) were recorded prior to imaging. Total length of fish from Gunderson Ditch was measured in the field prior to freezing, and fish were fully thawed before ultrasound examinations occurred. During examination, the fish was submerged in a tub of water sourced from flow-through holding tanks (hatchery fish), fresh stream water (wild fish), or tap water (lethally sampled fish). Sedatives were not added to water in the examination tub, which allowed live fish to begin recovery from sedation during the exam.

All fish were imaged with an Ibex EVO® veterinary ultrasound machine and an L14X high frequency transducer (E. I. Medical Imaging). Ultrasound settings ranged from a frequency of 10–14 MHz, read depth of 2–4 cm, and gain of 26–46 dB. The frequency and read depth varied based on fish size, with a higher frequency and shallower read depth for smaller fish. The best images were typically produced with gain settings > 35 dB. For each survey, we saved photos (.jpg files) and/or 4–6 second videos (.avi files) for post-survey review.

Each ultrasound exam required two technicians. The first technician handled the fish and performed the ultrasound scan while the second technician recorded images, length and PIT tag data. To collect the ultrasound image, the first technician held the fish on its side just below water's surface and scanned the fish by holding the transducer parallel to the lateral line and posterior to the operculum. Because fish were submerged during examination, no added transmission medium (e.g., gel) was necessary to obtain ultrasound images. To orient the image, the technician first located the stomach (Fig 2) and then slowly moved the transducer up and down between the lateral line and the ventral side of the fish to visualize oocytes (if present). In larger mature females, oocytes were easy to identify and tended to fill the abdominal cavity (Fig 3a). In smaller mature females, oocytes were more difficult to identify and were typically located posterior or ventral to the stomach (Fig 3b).

Exams typically lasted 20–60 seconds per fish, with shorter durations as technicians became more adept at obtaining high-quality images and identifying oocytes. Technicians discussed images and came to a consensus on the presence of oocytes in an individual before beginning the next exam. Fish were categorized as either maturing female or immature/male. Live fish were immediately placed in a recovery bucket with fresh hatchery or stream water.

We did not attempt to distinguish between immature and mature males because this information was not directly related to our objectives. Furthermore, identification of males with ultrasound tends to be less accurate than for females, largely because maturing ovaries are much easier to distinguish from other organs [26,28,42]. However, we did note fish presenting clear secondary sex characteristics (such as the presence of a kype). We did not attempt to estimate fecundity of maturing females because we did not have an opportunity to perform egg counts in non-lethally sampled populations. However, we were able to collect this information for maturing females from the Gunderson Ditch population that were lethally sampled on 5 November 2018 (n = 8). For each mature female, we estimated fecundity by weighing both ovaries, and then weighing a subset of 30–50 oocytes. We estimated the weight per oocyte by dividing the known number of oocytes in the subset by the total weight of the subset. We then divided the combined weight of both ovaries by the estimated weight per oocyte to obtain the number of oocytes. To estimate oocyte size, we placed 10 oocytes from a mature female next to

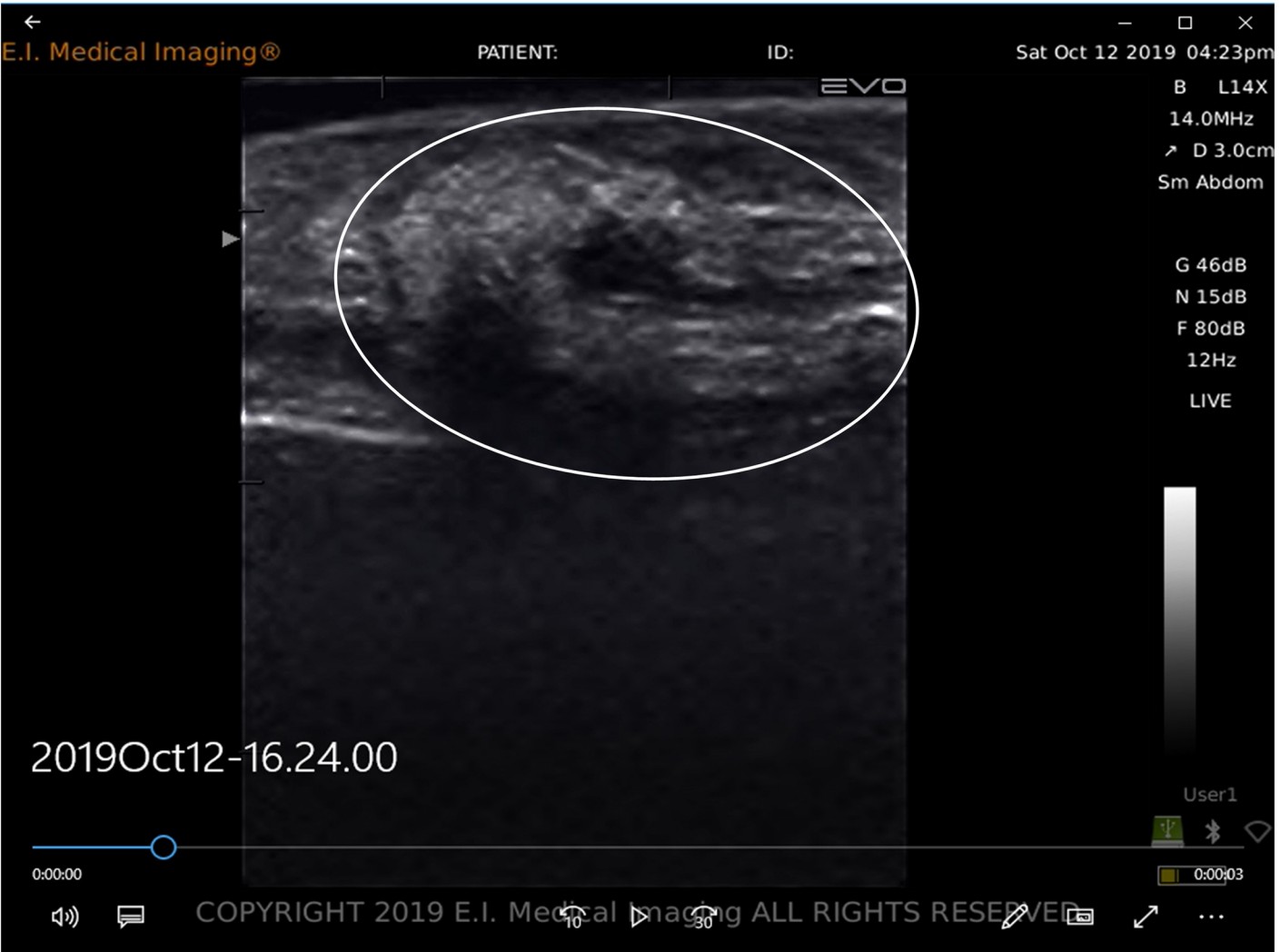

**Fig 2. Ultrasound image showing the stomach as a U-shaped object (circled).**

one another on a ruler, measuring the cumulative diameter of those oocytes, and then divided the cumulative diameter by 10. This value was then averaged across all maturing females to obtain an average oocyte size for the population.

### Data analysis

To determine the accuracy of ultrasound imaging, we compared the results of ultrasound exams to assessment of hatchery-reared fishes during and after spawning or euthanized fish assessed with dissection. The false negative rate was defined as the number of examinations wherein a maturing female was misclassified as a male or immature, divided by the total number of examinations. The false positive rate was defined as the number of examinations wherein a male or immature fish was misclassified as a maturing female, divided by the total number of examinations. The overall error rate was defined as the total number of misclassifications over the total number of examinations. We calculated the false negative, false positive and overall error rates for each of the six surveys at Sekokini Springs Hatchery and the

(a)

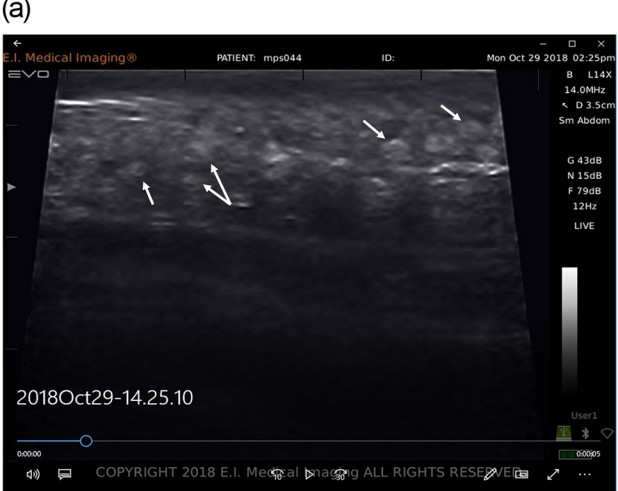

(b)

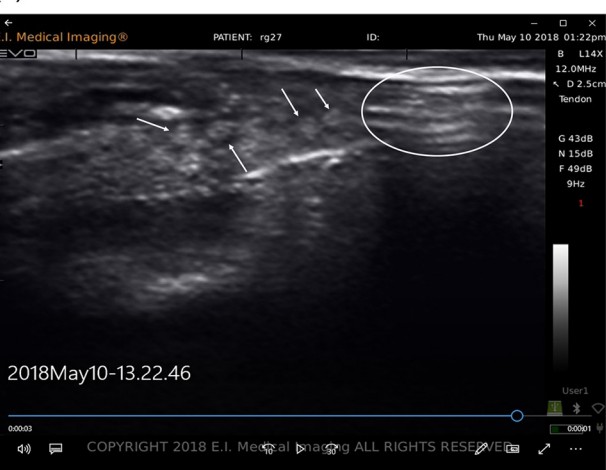

(c)

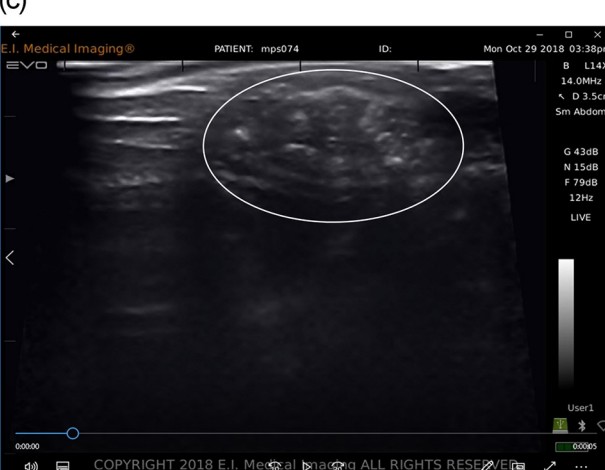

**Fig 3. Ultrasound images of mature female fish.** a) shows easily identifiable eggs in a 250 mm TL female from Magpie Spring Creek, MT imaged in October 2018. Eggs appear as soft, round orbs and fill the abdominal cavity. Arrows point to several of many eggs. b) A mature female at 121 mm TL collected from Gunderson Ditch, MT in November 2017. The white circle shows the stomach and pyloric caeca (horizontal parallel lines); arrows point to eggs. c) A fish at 108 mm TL from Magpie Spring Creek, MT. The circle shows the stomach with contents that resemble eggs. Dissection revealed this fish was a mature male with a full stomach. Unlike eggs in a and b, the round objects in this image are confined to the stomach, are more irregularly shaped, and have a more speckled appearance.

examination of fish from Gunderson Ditch. Hatchery fish that were surveyed more than once were also assessed for consistency of classifications across surveys.

We performed a Kruskal-Wallis test to quantify differences in the median length of mature females sourced from wild populations, including Cooney, Magpie Spring, Schley, South Fork Davis and Yellow Bay Creeks. A Kruskal-Wallis test was chosen because it does not assume a normal distribution of residuals [43]. We performed a Dunn post-hoc test to look for pairwise differences among populations. A Dunn post-hoc test was chosen because it does not assume equal sample sizes between groups. Gunderson Ditch was not included in this analysis because the field crews focused effort on collection of fish under 150 mm TL, and the sample was therefore biased. Significance was assessed at $\alpha \leq 0.05$.

## Results

### Accuracy of ultrasound methods

Ultrasound imaging was accurate at identifying mature female Cutthroat Trout. Over the course of six surveys at Sekokini Springs Hatchery, we performed 799 examinations on 543 unique fish ranging from 113–405 mm TL (mean = 243 mm). Misclassifications occurred in four of 799 examinations, for an overall error rate of 0.5% (Table 1). All four misclassification were false positives (misclassifying a fish as a maturing female). We examined 51 individuals lethally sampled from Gunderson Ditch that ranged from 103–171 mm TL (mean = 125 mm). The overall error rate for fish from Gunderson Ditch was 2.0%, resulting from one single false positive classification and one false negative classification. The fecundity of eight females collected from Gunderson Ditch in November 2018 ranged from 33–126 eggs per fish. Average egg diameter across eight maturing females collected from Gunderson Ditch in November 2018 ranged from 1.67–2.85 mm. One female at 106 mm TL collected in November 2018 was classified as immature/male, but dissection revealed ovaries with smaller, misshapen oocytes.

**Table 1. Accuracy of ultrasound examinations for identifying mature female Westslope Cutthroat Trout in resident populations from western Montana.**

| Source Population | Date | Sample Size | Size Range (mm) | # Mature Females | Mature Female Size Range (mm) | Overall Error Rate | False Positive Rate |
|---|---|---|---|---|---|---|---|
| Hatchery—Spawn Year 2018 | | | | | | | |
| Danaher | 11/21/2017 | 23 | 154–348 | 1 | 306 | 0% | 0% |
| Sullivan | 11/21/2017 | 26 | 141–256 | 8 | 147–221 | 0% | 0% |
| Young's | 11/21/2017 | 4 | 203–250 | 4 | 203–250 | 0% | 0% |
| Danaher | 1/16/2018 | 33 | 201–339 | 5 | 223–331 | 3.0% (1) | 3.0% (1) |
| Sullivan | 1/16/2018 | 50 | 152–330 | 20 | 157–330 | 4.0% (2) | 4.0% (2) |
| Young's | 1/16/2018 | 5 | 198–260 | 4 | 198–260 | 0% | 0% |
| Danaher | 3/6/2018 | 54 | 170–360 | 4 | 228–320 | 0% | 0% |
| Sullivan | 3/6/2018 | 56 | 160–288 | 23 | 160–233 | 0% | 0% |
| Young's | 3/6/2018 | 8 | 200–271 | 5 | 200–271 | 0% | 0% |
| **TOTAL** | | **259** | **141–360** | **74** | **147–330** | **1.2% (3)** | **1.2% (3)** |
| Hatchery—Spawn Year 2019 | | | | | | | |
| Danaher | 10/25/2018 | 77 | 141–363 | 10 | 308–355 | 0% | 0% |
| Sullivan | 10/25/2018 | 122 | 113–388 | 24 | 199–388 | 0.8% (1) | 0.8% (1) |
| Danaher | 1/22/2019 | 86 | 163–390 | 28 | 234–386 | 0% | 0% |
| Sullivan | 1/22/2019 | 86 | 130–392 | 20 | 223–392 | 0% | 0% |
| Danaher | 3/13/2019 | 105 | 169–405 | 30 | 234–395 | 0% | 0% |
| Sullivan | 3/13/2019 | 64 | 174–321 | 29 | 205–321 | 0% | 0% |
| **TOTAL** | | **540** | **113–405** | **141** | **199–308** | **0.7% (1)** | **0.7% (1)** |
| Euthanized Fish | | | | | | | |
| Gunderson Ditch | 11/16/2017; 11/20/2017 | 30 | 105–171 | 18 | 118–171 | 0% | 0% |
| Gunderson Ditch | 11/2018 | 21 | 103–146 | 8 | 109–146 | 4.8% (1) | 4.8% (1) |
| **TOTAL** | | **51** | **103–171** | **26** | **109–171** | **2.0% (1)** | **2.0% (1)** |

"Date" refers to the date of the ultrasound survey conducted at Sekokini Springs Hatchery, or date of capture for euthanized fish from Gunderson Ditch. Note that the samples size of fish from Sekokini Springs Hatchery includes repeated observations of individuals over time. For overall error (number of misclassifications divided by total number of fish examined) and false positive error rates (number of fish misclassified as maturing female over total number of fish examined), the number of misclassifications is listed in parentheses. No false negative errors (maturing females misclassified as male or immature) were observed in this dataset.

**Table 2. Sampling details and ultrasound examination results for wild Westslope Cutthroat Trout populations in western Montana surveyed with ultrasound.**

| Population | Survey Dates | Sample Size | # Mature Females | Length of Mature Females (mm) | |
|---|---|---|---|---|---|
| | | | | Range | Median |
| Gunderson Ditch | 11/16/2017; 11/20/2017; 11/5/2018 | 51 | 26 | 109–171 | N/A |
| Cooney | 10/15/2018; 10/17/2018 | 77 | 15 | 147–235 | 183 |
| Magpie Spring | 10/29/2018 | 66 | 28 | 124–250 | 185.5 |
| Schley | 10/1/2019 | 34 | 11 | 130–180 | 166 |
| South Fork Davis | 10/12/2019 | 23 | 9 | 141–180 | 148 |
| Yellow Bay | 10/2/2019 | 17 | 4 | 150–204 | 164 |

Fish collection in Gunderson Ditch generally focused on individuals under 150 mm in total length. This was not representative of the entire adult populations and therefore the median size of mature females was not calculated for this population.

A subset of the ten most intact oocytes from this fish had an average diameter of 1.43 mm. In contrast, oocyte diameter across the eight maturing females ranged from 1.67–2.85 mm, with an overall average of 2.18 mm for all eight females. Due to the small size and poor condition of eggs in the 106 mm TL female, we concluded that her eggs were likely not viable and the classification of this fish as immature was accurate [see 21, 40].

Results of ultrasound examinations of hatchery-reared populations were consistent across time and effective at identifying mature females for up to eight months prior to spawning. Of the 252 individuals examined during fall hatchery surveys (November 2017 and October 2018), only one fish sampled in October 2018 was misclassified as a maturing female. This fish was reexamined in January and March of 2019 and was correctly classified as male/immature. Furthermore, repeated examinations of individuals prior to spawning were consistent through time. A total of 211 hatchery fish were examined on multiple visits within a spawning year, 45 of which were examined at all three visits prior to spawning season (S1 Table). Classifications of all fish were consistent between repeated examinations except for the male fish misclassified as maturing female in October 2018 (see above).

### Variation in size of maturing Cutthroat Trout

The average size at maturity varied significantly among wild populations ($\chi^2$ = 14.83, $p$ = 0.01, df = 4; Table 2, Fig 4, S1–S6 Figs). The median size of maturing females was largest in Magpie Spring Creek (186 mm TL) and smallest in South Fork Davis Creek (166 mm TL), with significant differences between South Fork Davis Creek and Cooney Creek ($p < 0.01$), and between South Fork Davis Creek and Magpie Spring Creek ($p$ = 0.03). The size of the smallest maturing female in each population ranged from 109 mm TL in Gunderson Ditch to 150 mm TL in Yellow Bay Creek (Table 2). Fish in Gunderson Ditch ranged from age 1 to 7 (S7 Fig); the smallest mature female at 109 mm TL was 3 years old.

### Discussion

This study demonstrates that ultrasound imaging can be used to accurately identify maturing female Westslope Cutthroat Trout as small as 109 mm TL and up to eight months prior to spawning. For this species, gonad development more than six months prior to spawning in late spring may be adaptive, resulting from their specific habitat requirements and life history strategy. Resident Cutthroat Trout populations typically occur in high-elevation streams with short growing seasons lasting only 2–3 months. To ensure adequate energy investment in gametes, this species may begin gonad development in late summer and fall when food availability and metabolic rates are higher. For example, Foster (40) found that gonads of adult Coastal

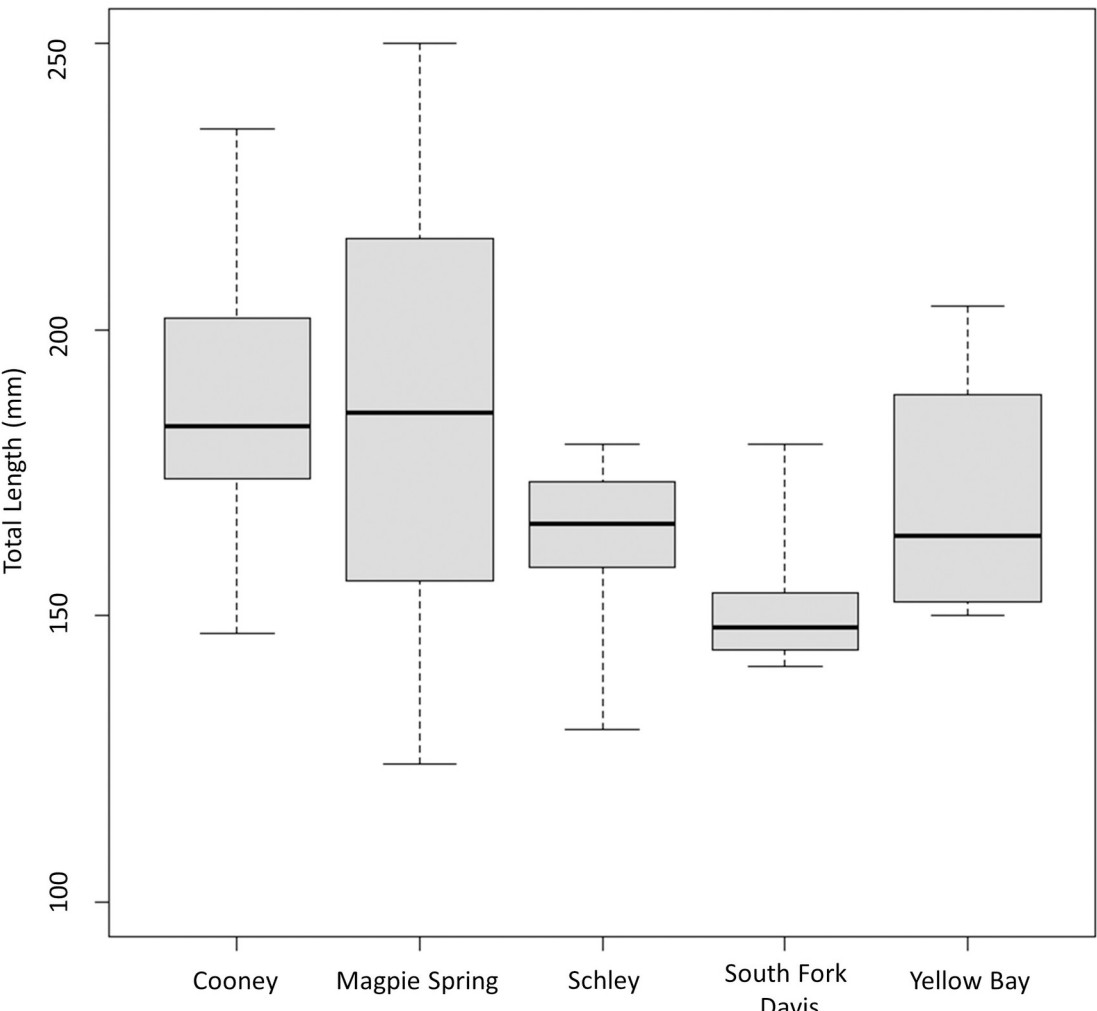

**Fig 4. Box plot showing the total length (mm) of mature female Westslope Cutthroat Trout from wild resident populations in western Montana.** Bold black lines are the median total length of mature females, boxes show the interquartile range, and whiskers are the data range. A Dunn's post-hoc test found significant differences in the median size of maturing females between South Fork Davis Creek and both Cooney and Magpie Spring Creeks, respectively.

Cutthroat Trout (*O. c. clarkii*) could not be identified via dissection in July or August, but development was evident by September, approximately seven to eight months prior to spawning. By October, females could be easily categorized into two groups- those with "large" ovaries containing distinct oocytes typically over 2 mm in diameter, and females with substantially smaller ovaries containing poorly developed, minuscule oocytes. The authors categorized the fish with "large ovaries" as maturing females, and those with "small ovaries" as immature, hypothesizing that they were unlikely to spawn the following spring. In the same trout population, Bangs and Nagler (21) demonstrated that levels of vitellogenin and estradiol 17-β in blood plasma measured in October and April (less than one month prior to spawning) were significantly higher in females with "large" ovaries compared to those with "small" ovaries. The increased levels of vitellogenin and estradiol in females with "large" ovaries further supports the classification of these individuals as mature. In addition, Bangs and Nagler (21) demonstrated that ultrasound imaging could be used to identify females with "large" ovaries in

April, and anecdotal evidence suggested this tool could be used to identify maturing females as early as October. Building upon these studies, we coupled information from ultrasound examinations of hatchery-reared individuals with spawn and post-spawn inventories to verify that females with substantial oocyte development in October reached maturity the following spring. This verification, in addition to accurate ultrasound examinations of Gunderson Ditch fish that were lethally sampled in November, validated the use of ultrasound as a tool for distinguishing maturing females months prior to spawning. We were therefore confident in our ability to non-lethally assess maturity of wild Westslope Cutthroat Trout populations in the fall when streams were more accessible and spawning behavior would not be disturbed.

## Accuracy of ultrasound methods

While validating these methods, we identified several techniques that aided accuracy of ultrasound assessments. Recording ultrasound examinations as short (4–6 sec) videos provided a more detailed record than still images. This increased detail was useful for reviewing uncertain classifications. For example, after we had validated our ultrasound protocol, we reviewed videos of the three misclassified fish surveyed in January 2018 (Table 1). We concluded that we would now (accurately) classify all three fish as immature/male. The practice of reviewing videos to ensure accurate assessments can be particularly helpful when imaging in the field, where variable conditions and time constraints can make classifications more difficult. Furthermore, beginning a field survey, we found it useful for technicians to reestablish a search image by reviewing videos of difficult classifications from previous surveys. During surveys of wild populations, we also learned that smaller fish with full stomachs were more difficult to distinguish from fish with eggs and could lead to false positive misclassifications. Under these circumstances, we learned to distinguish eggs as dull, round orbs, whereas full stomachs tended to have a sharper, more speckled and irregular appearance (e.g., Fig 3c). In future wild fish surveys, purging gut contents with gastric lavage may help minimize false positive classifications, especially for small fish.

Consistent with other studies, we found that developing oocytes were very easy to identify in individuals larger than 120 mm TL due to larger gonads and higher fecundities [see 26]. Here, ovaries generally filled the abdominal cavity and oocytes were easily identified in the ultrasound image (Fig 3a). Oocytes were more difficult to identify in smaller fish (< 120 mm TL) because fecundities tended to be low. In these cases, we observed that ovaries were generally situated either posterior or ventral to the stomach (Fig 3b). Understanding the location of ovaries, given the image orientation, increased confidence in the identification of oocytes in smaller fish. While we did not observe any false negative classifications of maturing female fish in our study, these may still have occurred during surveys of wild populations where assessments were not validated with a secondary method.

Our study also adds to a small body of work exploring the use of ultrasound to assess reproductive status in small-bodied fishes and at various time points prior to spawning. Only one other study has used ultrasound to identify mature female fish smaller than 200 mm TL. Bryan, Wildhaber [44] identified mature female Neosho Madtoms (*Noturus placidus*) ranging from 80–120 mm TL based on the presence of abundant ova in ultrasound images collected during spawning season. An important consideration, evident in the literature, is that the timeframe for assessing sex and reproductive status with ultrasound methods prior to spawning varies by species. For example, egg development in Red Hind (*Epinephelus guttatus*) typically begins 3 to 4 months prior to mating [45]. However, Whiteman, Jennings [46] found that sex determination of mature Red Hind with ultrasound was not possible more than one month prior to spawning because gonads were not yet large enough to be distinguished from

other organs. In contrast, Næve, Mommens [47], reliably identified eggs of hatchery reared Atlantic Salmon *(Salmo salar)* using ultrasound approximately 4 months prior to fall spawning. These studies suggest that ultrasound methods may be used to assess sex and reproductive status of fishes more broadly than they are currently being applied. However, methods must be validated to determine when accurate information may be obtained within a species' reproductive cycle.

### Variation in Cutthroat Trout size at maturity

Similar to studies on other salmonids, [13–15,17,18], we observed significant variation in size of maturing females across six resident Westslope Cutthroat Trout populations. Furthermore, in Gunderson Ditch we observed multiple maturing females under 120 mm TL (S1 Fig), approximately 20 mm smaller than has been previously reported for resident populations of this species [37,38]. Size at maturity across populations of salmonids varies with numerous environmental factors [13,15–19]. Research demonstrates that this trait is phenotypically plastic [7,48,49]. Plasticity may favor persistence of salmonids as they encounter environmental changes such as increased stream temperatures and more variable flow regimes associated with climate change [50]. Additionally, plasticity in the minimum size at maturity may help promote persistence of small isolated salmonid populations. For example, Carim, Vindenes (10) found that models of population growth rates for isolated resident Westslope Cutthroat Trout in western Montana were highly sensitive to the size of mature females. Specifically, a small decrease (<5% change) in the minimum size of the mature females resulted in a large increase in a population's growth rate. The sensitivity of these models to the size of maturing females was highest for populations with less than 1 km of occupied habitat. Unfortunately, the authors were unable to directly measure size at maturity in their study populations, and instead used generalized estimates from similar populations in the region [37]. This resulted in remarkably low estimates of population persistence, despite little to no change in population densities over several decades prior to this study.

Although multiple lines of evidence suggest that size at maturity varies across Westslope Cutthroat Trout populations, the minimum age at maturity appears to be more consistent. In resident populations, the youngest mature female documented in published studies has consistently been three years old [37,38]. Previously developed length-at-age keys for the wild populations in our study (Cooney, Magpie Spring, Schley, South Fork Davis and Yellow Bay) indicate that the smallest maturing female observed across these populations was three years old [10; K. Carim unpublished data]. And similarly, direct aging of otoliths of fish from Gunderson Ditch indicates that the smallest mature female at 109 mm was also three years old. It is important to note these data do not suggest all females of this species mature at age-3. However, these data do suggest that females of this species may require some minimum number of years to reach maturity, independent of size. This is particularly important as we consider the ability of populations to adapt to isolation in harsh environments where growth is limited, and survival is likely low.

### Conclusions

Ultrasound technology has been used for over 35 years to assess sex and reproductive maturity across dozens of fish species [26]. In this study, we demonstrate that ultrasound imaging may further be used to assess reproductive maturity in small-bodies salmonids up to eight months prior to spawning. The ability of this tool to accurately and non-invasively identify maturing females outside the breeding season will allow researchers to more precisely measure variation in size of maturity among populations of vulnerable salmonid populations. This information

may be incorporated into viability models [10,11,34–36] to better understand how this trait influences population growth rates and persistence.

## Supporting information

**S1 Fig. Length-frequency distribution of fish in wild, resident populations surveyed in the seven to eight months prior to spawning.** Light grey bars represent the frequency of all fish sampled in a given size range; dark bars represent the frequency of fish identified as maturing females. Note that sampling in Gunderson Ditch targeted fish between 100–150mm total length. Sampling in other populations did not target fish of any particular size range. (DOCX)

**S2 Fig. Length-frequency distribution of fish in wild, resident populations surveyed in the seven to eight months prior to spawning.** Light grey bars represent the frequency of all fish sampled in a given size range; dark bars represent the frequency of fish identified as maturing females. Note that sampling in Gunderson Ditch targeted fish between 100–150mm total length. Sampling in other populations did not target fish of any particular size range. (DOCX)

**S3 Fig. Length-frequency distribution of fish in wild, resident populations surveyed in the seven to eight months prior to spawning.** Light grey bars represent the frequency of all fish sampled in a given size range; dark bars represent the frequency of fish identified as maturing females. Note that sampling in Gunderson Ditch targeted fish between 100–150mm total length. Sampling in other populations did not target fish of any particular size range. (DOCX)

**S4 Fig. Length-frequency distribution of fish in wild, resident populations surveyed in the seven to eight months prior to spawning.** Light grey bars represent the frequency of all fish sampled in a given size range; dark bars represent the frequency of fish identified as maturing females. Note that sampling in Gunderson Ditch targeted fish between 100–150mm total length. Sampling in other populations did not target fish of any particular size range. (DOCX)

**S5 Fig. Length-frequency distribution of fish in wild, resident populations surveyed in the seven to eight months prior to spawning.** Light grey bars represent the frequency of all fish sampled in a given size range; dark bars represent the frequency of fish identified as maturing females. Note that sampling in Gunderson Ditch targeted fish between 100–150mm total length. Sampling in other populations did not target fish of any particular size range. (DOCX)

**S6 Fig. Length-frequency distribution of fish in wild, resident populations surveyed in the seven to eight months prior to spawning.** Light grey bars represent the frequency of all fish sampled in a given size range; dark bars represent the frequency of fish identified as maturing females. Note that sampling in Gunderson Ditch targeted fish between 100–150mm total length. Sampling in other populations did not target fish of any particular size range. (DOCX)

**S7 Fig. Length vs. age of fish captured in Gunderson Ditch.** Filled circles represent females with eggs; open circles reprent all other fish. Several age-1 fishe were larger than expected for this populations. Fish sampled from Gunderson Ditch likley may represent a mix of individuals spanwned and reared in upstream areas, as well as some spawned and reared within the ditch. Differences in rearing conditions (e.g., temperature, food avauilability, etc.) between the

ditch and upstream areas could explain the broad variaion in size of in age-1 fish.
(DOCX)

**S1 Table. Consistency in maturity classifications of individual fish at Sekokini Springs Hatchery over repeated examinations.**
(DOCX)

## Acknowledgments

We would like to acknowledge the people of the Salish and Kootenai Tribes and their homelands where we conducted this study. We would like to thank Jim Dunnigan, Toby Tabor, Sheldon Fisher, Chris Grenier, Bruce Maestas, Coby Roberts, Joe Santos, Mountain Wahl, Karen Lohman, Tanner Banks, Troy Smith, and Vickie Zeller for their assistance in sample and data collection. We also thank Katie Zarn and Rebecca Smith for their comments on earlier drafts of this manuscript.

## Author Contributions

**Conceptualization:** Kellie J. Carim, Andrew R. Whiteley.

**Data curation:** Kellie J. Carim, Scott Relyea, John A. Kronenberger.

**Formal analysis:** Kellie J. Carim.

**Funding acquisition:** Kellie J. Carim, Beau Larkin.

**Investigation:** Kellie J. Carim, Scott Relyea, Craig Barfoot, Lisa A. Eby, John A. Kronenberger, Beau Larkin.

**Methodology:** Kellie J. Carim, Lisa A. Eby.

**Project administration:** Kellie J. Carim.

**Resources:** Kellie J. Carim.

**Writing – original draft:** Kellie J. Carim.

**Writing – review & editing:** Kellie J. Carim, Scott Relyea, Craig Barfoot, Lisa A. Eby, John A. Kronenberger, Andrew R. Whiteley, Beau Larkin.

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
