## [Decision Letter · Decision Letter 0]

17 Dec 2020

PONE-D-20-26995

Ultrasound imaging identifies life history variation in resident Cutthroat Trout

PLOS ONE

Dear Dr. Carim,

Thank you for submitting your manuscript to PLOS ONE. After careful consideration, we feel that it has merit but does not fully meet PLOS ONE’s publication criteria as it currently stands. Therefore, we invite you to submit a revised version of the manuscript that addresses the points raised during the review process.

The manuscript is well written and the data and subsequent analyses are sound. The information is of interest to those studying cutthroat trout and other NA fish. Two reviewers had only minor editorial comments. However, these changes are necessary and require the author's attention.  The authors should be complemented on a well put together submission.  

We look forward to receiving your revised manuscript.

Kind regards,

Madison Powell, PhD

Academic Editor

PLOS ONE

Additional Editor Comments:

The manuscript is well written and the data and subsequent analyses are sound. Two reviewers had only minor editorial comments. However, these changes are necessary and require the author's attention. The authors should be complemented on a well put together submission.

2. We note that Figure 1 in your submission contains map images which may be copyrighted. All PLOS content is published under the Creative Commons Attribution License (CC BY 4.0), which means that the manuscript, images, and Supporting Information files will be freely available online, and any third party is permitted to access, download, copy, distribute, and use these materials in any way, even commercially, with proper attribution. For these reasons, we cannot publish previously copyrighted maps or satellite images created using proprietary data, such as Google software (Google Maps, Street View, and Earth). For more information, see our copyright guidelines: http://journals.plos.org/plosone/s/licenses-and-copyright.

(1) You may seek permission from the original copyright holder of Figure 1 to publish the content specifically under the CC BY 4.0 license. 

Reviewers' comments:

Reviewer's Responses to Questions

**Comments to the Author**

1. Is the manuscript technically sound, and do the data support the conclusions?

Reviewer #1: Yes

Reviewer #2: Yes

2. Has the statistical analysis been performed appropriately and rigorously? 

Reviewer #1: Yes

Reviewer #2: Yes

3. Have the authors made all data underlying the findings in their manuscript fully available?

Reviewer #1: Yes

Reviewer #2: Yes

4. Is the manuscript presented in an intelligible fashion and written in standard English?

Reviewer #1: Yes

Reviewer #2: Yes

5. Review Comments to the Author

Reviewer #1: This paper describes the testing and validation of ultrasound imaging for assessing the reproductive maturity of female westslope cutthroat trout. Validation came from using known-age fish reared at a hatchery and wild fish, lethally sampled and dissected following imaging. The major findings of the study were that ultrasound imaging accurately identified maturing female cutthroat trout of small size and up to 8 months prior to spawning. They additionally showed large population variation in size of maturing females. I thought the paper is generally well written and highlights a tool that can be used in settings where biologists want to avoid lethal sampling. My only two criticisms of the study is that they didn’t try a genetic sex marker which would have allowed them to differentiate immature males from immature females, and it wasn’t clear why they didn’t report otolith ages of fish they sampled from Gunderson Ditch. They mention that they did do direct aging, but offer no methods or results. I liked the discussion they provided on techniques that aided in ultrasound assessments.

I only have a few additional minor edits/comments below:

Line 276: Isn’t Kruska-Wallis for ranked variables?

Line 346: “as small” is repeated

Lines 431-439: This paragraph would also work well in the introduction when you are making the case for needing better data on size at maturity.

Reviewer #2: Your paper presents novel research that is widely applicable to conservation of cutthroat trout and presumably to other salmonids, assuming that your methods apply to similar species. I found all aspects of the research to be methodologically sound, although my technical expertise relevant to this paper is limited to statistics and population viability analysis. Other reviewers will need to comment on the ultrasound methods. The paper is one of the clearest, most concise, appropriately documented (literature review and supporting data), and well written manuscripts I have reviewed in many years. Conclusions are sound and are supported by the data, in context of appropriate literature. Thank you for such careful attention to conducting and reporting this work.

I found four very minor grammatical/spelling issues.

Line 63: insert the word "of" between "persistence" and "populations".

Line 348: insert the word "be" between "may" and "adaptive".

Line 529: second word in article title is "identification".

Line 530: last word in journal title is "Embryology".

6. PLOS authors have the option to publish the peer review history of their article (what does this mean?). If published, this will include your full peer review and any attached files.

Reviewer #1: No

Reviewer #2: **Yes: **Robert W. Van Kirk

---

## [Author Response · Author response to Decision Letter 0]

23 Dec 2020

Additional Editor Comments:

The manuscript is well written and the data and subsequent analyses are sound. Two reviewers had only minor editorial comments. However, these changes are necessary and require the author's attention. The authors should be complemented on a well put together submission.

Journal requirements: When submitting your revision, we need you to address these additional requirements.

KC: We updated the style and format for the cove page, headers, figure legends, and table legends. We also are submitting figures individually as .tiff files with the proper naming convention. 

2. We note that Figure 1 in your submission contains map images which may be copyrighted. All PLOS content is published under the Creative Commons Attribution License (CC BY 4.0), which means that the manuscript, images, and Supporting Information files will be freely available online, and any third party is permitted to access, download, copy, distribute, and use these materials in any way, even commercially, with proper attribution. For these reasons, we cannot publish previously copyrighted maps or satellite images created using proprietary data, such as Google software (Google Maps, Street View, and Earth). For more information, see our copyright guidelines: http://journals.plos.org/plosone/s/licenses-and-copyright.

KC: The base map in this Figure 1 is the “Terrain with Labels” base map that is a publicly available in ArcGIS software. In the original figure, the placement of the inset map covered the base map copyright information. I’ve updated Figure 1. To ensure that the copyright information is displayed in the lower right-hand corner. Based on the sources below, this should be sufficient for citing this base map layer. 

https://support.esri.com/en/technical-article/000012040

https://community.esri.com/t5/arcgis-online-questions/citation-information-for-imagery-basemaps-in-arcgis-online/m-p/484078

KC: We added the caption for supporting information to the end of the manuscript document and have updated the in-text citations to match the journal guidelines. 

Reviewers' Comments to the Author

Reviewer #1: This paper describes the testing and validation of ultrasound imaging for assessing the reproductive maturity of female westslope cutthroat trout. Validation came from using known-age fish reared at a hatchery and wild fish, lethally sampled and dissected following imaging. The major findings of the study were that ultrasound imaging accurately identified maturing female cutthroat trout of small size and up to 8 months prior to spawning. They additionally showed large population variation in size of maturing females. I thought the paper is generally well written and highlights a tool that can be used in settings where biologists want to avoid lethal sampling. My only two criticisms of the study is that they didn’t try a genetic sex marker which would have allowed them to differentiate immature males from immature females, and it wasn’t clear why they didn’t report otolith ages of fish they sampled from Gunderson Ditch. They mention that they did do direct aging, but offer no methods or results. I liked the discussion they provided on techniques that aided in ultrasound assessments.

KC: We agree that a genetic marker to identify sex of immature fish would be extremely helpful. Colleagues in the Genetics Program at Idaho Fish and Game (IDFG) are working on developing a sex marker for this species. We sent our colleagues genetic samples from mature fish in this study to test the accuracy their sex identification marker in our populations. At that time, the assay did not perform with high enough accuracy for us to incorporate the marker into our study. However, we are hopeful that our samples may be used to develop a more accurate marke. We have added text to lines 168-170 to clarify that a sex marker is not available. 

Thank you for pointing out the lack of information on how otoliths were used to age fish as well as reporting of the results. We have added text (lines 193 – 196 and 342-343) describing the methods and results of aging, as well as a supplementary figure (S7) showing the length and corresponding age of fish from Gunderson Ditch.

I only have a few additional minor edits/comments below:

Line 276: Isn’t Kruska-Wallis for ranked variables?

KC: The test is not only for ranked variables. It broadly is a test that can be used to compare differences in a continous dependent variable (in this case, fish length) by a categorical independent variable (in this case, population). This test is used as an alternative to an ANOVA when the assumptions of an ANOVA (e.g., the dependent variable is normally distributed) are not met. Further examples may be found in the references below. Based on these references, and because Review #2 has a strong background in statistics and did not comment on the use of this test, we feel we used it appropriately. 

https://www.statisticssolutions.com/kruskal-wallis-test/

https://en.wikipedia.org/wiki/Kruskal%E2%80%93Wallis_one-way_analysis_of_variance#cite_note-Laerd-1

Line 346: “as small” is repeated

KC: We deleted the repeated text. Thank you!

Lines 431-439: This paragraph would also work well in the introduction when you are making the case for needing better data on size at maturity.

KC: Thank you for this suggestion. We agree that this example provides justification for the work in this study. We worked to keep the introduction broad in order to appeal to a wider audience. We felt this detailed example was too narrow to be included in the introduction. Instead, we have chosen to include this text in the discussion to help foreshadow how we plan to build upon the work presented in this manuscript. 

Reviewer #2: Your paper presents novel research that is widely applicable to conservation of cutthroat trout and presumably to other salmonids, assuming that your methods apply to similar species. I found all aspects of the research to be methodologically sound, although my technical expertise relevant to this paper is limited to statistics and population viability analysis. Other reviewers will need to comment on the ultrasound methods. The paper is one of the clearest, most concise, appropriately documented (literature review and supporting data), and well written manuscripts I have reviewed in many years. Conclusions are sound and are supported by the data, in context of appropriate literature. Thank you for such careful attention to conducting and reporting this work.

KC: We are flattered by the compliments provided by both Reviewers 1 and 2. Thank you for the kind and supportive comments! It’s always nice to have some positive feedback with the critique. 

I found four very minor grammatical/spelling issues.

Line 63: insert the word "of" between "persistence" and "populations".

KC: Done. 

Line 348: insert the word "be" between "may" and "adaptive".

KC: Done. 

Line 529: second word in article title is "identification".

KC: Corrected. 

Line 530: last word in journal title is "Embryology".

KC: Corrected. 

KC: We used PACE to ensure that all figures meet PLOS requirements. (I have never used this tool before. I found it very useful and user friendly!)

---

## [Editor Report · Decision Letter 1]

19 Jan 2021

Ultrasound imaging identifies life history variation in resident Cutthroat Trout

PONE-D-20-26995R1

Dear Dr. Carim,

We’re pleased to inform you that your manuscript has been judged scientifically suitable for publication and will be formally accepted for publication once it meets all outstanding technical requirements.

Kind regards,

Madison Powell, PhD

Academic Editor

PLOS ONE

Additional Editor Comments (optional):

This manuscript has been resubmitted with all necessary changes indicated by the reviewers. The data are publishable in their present form.
---

## [Editor Report · Acceptance letter]

21 Jan 2021

PONE-D-20-26995R1 

Ultrasound imaging identifies life history variation in resident Cutthroat Trout 

Dear Dr. Carim:

I'm pleased to inform you that your manuscript has been deemed suitable for publication in PLOS ONE. Congratulations! Your manuscript is now with our production department. 

Kind regards, 

on behalf of

Dr. Madison Powell 

Academic Editor

PLOS ONE